# Enhanced Surface Energetics of CNT-Grafted Carbon Fibers for Superior Electrical and Mechanical Properties in CFRPs

**DOI:** 10.3390/polym12061432

**Published:** 2020-06-26

**Authors:** Arash Badakhsh, Kay-Hyeok An, Byung-Joo Kim

**Affiliations:** 1Center for Hydrogen Energy and Fuel Cell Research, Korea Institute of Science and Technology (KIST), Seoul 02792, Korea; badakhsh.arash@gmail.com; 2Department of Mechanical Design Engineering, Jeonbuk National University, Jeonju 54896, Korea; 3Department of Carbon and Nano Materials Engineering, Jeonju University, Jeonju 55069, Korea; 4Research Laboratory for Multifunctional Carbon Materials, Korea Institute of Carbon Convergence Technology (KCTECH), Jeonju 54853, Korea

**Keywords:** carbon nanotube, CFRP, surface energy, electrical properties, mechanical properties

## Abstract

Surface enhancement of components is vital for achieving superior properties in a composite system. In this study, carbon nanotubes (CNTs) were grown on carbon fiber (CF) substrates to improve the surface area and, in turn, increase the adhesion between epoxy-resin and CFs. Nickel (Ni) was used as the catalyst in CNT growth, and was coated on CF sheets via the electroplating method. Surface energetics of CNT-grown CFs and their work of adhesion with epoxy resin were measured. SEM and TEM were used to analyze the morphology of the samples. After the optimization of surface energetics by catalyst weight ratio (15 wt.% Ni), CF-reinforced plastic (CFRP) samples were prepared using the hand lay-up method. To validate the effect of chemical vapor deposition (CVD)-grown CNTs on CFRP properties, samples were also prepared where CNT powder was added to epoxy prior to reinforcement with Ni-coated CFs. CFRP specimens were tested to determine their electrical resistivity, flexural strength, and ductility index. The electrical resistivity of CNT-grown CFRP was found to be about 9 and 2.3 times lower than those of as-received CFRP and CNT-added Ni-CFRP, respectively. Flexural strength of CNT-grown Ni-CFRP was enhanced by 52.9% of that of as-received CFRP. Interestingly, the ductility index in CNT-grown Ni-CFRP was 40% lower than that of CNT-added Ni-CFRP. This was attributed to the tip-growth formation of CNTs and the breakage of Ni coating.

## 1. Introduction

Within the last few decades, composite systems have become the main focus of scientists and engineers in different fields, ranging from transportation [1] and building materials [2] to thermal management [3] and energy storage [4]. This is mainly due to the flexibility of composite materials that can be tailored for specific or multiple applications. A typical composite material consists of two main components, namely matrix and filler(s). Fillers are used to enhance the target properties in the matrix materials. Morphologically, fillers are mainly classified into particles and fibers. Fibrous systems benefit from extremely high mechanical strength and superior anisotropic thermal and electrical properties along the fiber axis [5]. Among the fibrous fillers, carbon fiber (CF) is of particular interest due to its exceptionally high strength-to-weight ratio, as well as superior electrical and thermal conductivities. The main challenge in advanced composite design is how to overcome the surface energy mismatch between the filler(s) and the matrix. One of the main trends in modern composite fabrication is the use of light-weight, mechanically tough, and cost-effective polymers, e.g., thermoset resins and thermoplastics, as the matrix. However, polymers generally have different surface energy characteristics than metals, ceramics, and carbon materials. Many have attempted to lower this energy mismatch by intensive surface treatments. For instance, Sun et al. [6] decorated carbon fibers with graphene oxide/polymer compound for better adhesion of fibers to the polymeric matrix, and improvement of mechanical properties in the composites. Shen et al. [7] employed carbon nanotubes (CNTs) along with a silane coupling agent to improve the mechanical properties of CF-reinforced plastic (CFRP) composites. They reported a 13.98% improvement in interlaminar shear strength (ILSS) of epoxy-based CFRPs. However, the chemical functionalization of CF has been shown to somehow damage the fiber surface by creating unfavorable sporadic pits [8]. Hence, we have opted for a more physical approach using adsorption of grafts on CF, in this case, linked by Van der Waals secondary force. We have shown in our previous studies that coating CF with adsorption of secondary agents can dramatically improve the CFRP properties. In our report published in Polymers, we concluded that modification of CF surface with boron nitride led to better interfacial adhesion of CF to the thermoplastic matrix, which in turn improved the thermal properties in composites [9]. In another study, we have optimized the effect of electrochemical etching time for the best CF performance under tensile load [10].

Nickel (Ni) has been reported to have one of the best catalytic activities for CNT growth [11]. Many studies have tried to analyze the effect of the catalyst on the quality of the grown CNTs. Wei et al. [12] concluded from their study on a silicon-based substrate that there was no strong correlation between the catalyst film thickness and the diameter of the tubes grown with conventional chemical vapor deposition (CVD). Atthipalli et al. [13] used sputtered Ni to facilitate the growth of CNTs on copper. They did not, however, discussed the effect of the catalyst on surface energetics of copper. The effect of Ni catalyst dispersion on the growth of carbon nanofibers on CF was studied by Meng et al. [14]. They used mesoporous silica as an interface between the CF substrate and the Ni catalyst. In their research, Du et al. [15] reported a well adhesion between the flame-assisted grown CNTs and CF substrate. They also analyzed the electrical and mechanical properties of the CNT-grafted CFRPs as a function of CNT growth time. However, they did not optimize the surface energetics of the CNT-grafted CFs for better composite properties. Hence, the enhancement achieved for the electrical properties was far (about 3-fold) less than the one obtained for the present study. Wettability of CFs before and after modification with CVD-grown CNTs at a very high temperature (1020 °C) were studied by Wang et al. [16]. They calculated the free surface energy of CNT-grafted CFs and compared it with Ni-coated and as-received CFs. Similar work was done by Singh Bedi et al. [17], where they analyzed the difference in epoxy-drop contact angle between the as-received and the CNT-grafted CFs. Guignier et al. [18] grew CNTs on CF fabrics using Fe_2_O_3_ and Ferrofluid as catalysts. The CNT-grafted CFs were characterized for their tribological and adhesion characteristics, as well as their wettability against epoxy. They concluded that the nature of the catalyst dramatically affected the properties of the modified surface. However, their experiments were focused on the chemical composition of the catalysts, rather than morphology, and wear properties of modified fabric for industrial-scale production. In another interesting study, CF tows were dip-coated in a ternary (Co, Ni, Fe) catalyst solution by Fan et al. [19]. They employed an electrochemical anodic oxidation method to observe the effect of treatment intensity on the CF surface properties. They reported the mechanical properties of fabricated CFRPs as a function of surface area, catalyst composition, and CVD temperature. Boroujeni and Al-Haik [20] reported a better (by 7%) tensile strength of CFRPs after patterned growth of CNTs on CF fabric compared with that of uniformly grown CNTs. However, unlike the present study, previous works have not discussed the effect of catalyst weight ratio on the surface energetics of CNT-grafted CFs and its effect on the enhancement of composite properties. Furthermore, to the best of our knowledge, ductility index values for composites reinforced with CNT-grafted CFs have not yet been reported.

The main contributions of this work are listed below:Optimization of catalyst weight on CF substrate for the highest surface energetics and crystallinity of grafting CNTs.Revealing the correlation between the crystallite size and surface energy of the as-grown carbon structures.Revealing the effect of optimized CNT-growth to achieve superior electrical and mechanical properties in CFRPs.Insights on the effect of CNTs as the secondary filler on the ductile response of respective CFRPs.

## 2. Experimental Details

### 2.1. Materials

CFs as the primary filler were fabric-type polyacrylonitrile (PAN)-based (T300, CO6644B) and purchased from Toray Industries Inc., Tokyo, Japan. The CF sheets were 300 g/m^2^ in weight, and each monofilament had an approximate diameter of 7 µm. Multi-walled CNT powder (95%< pure, mean diameter of 10 nm, mean length of 20 μm) for CNT-added CFRP fabrication was provided by Nanosolution Co., Jeonju-si, Korea. The epoxy matrix was diglycidyl ether bisphenol-A (DGEBA, YD-128) supplied by Kukdo Chemicals, Seoul, Korea. It had an epoxide equivalent weight of 187 g/eq and a viscosity of about 11,500–13,500 cPs at room temperature (RT). Diamineodiphenylmethane (DDM, Sigma-Aldrich Chemicals Co., St. Louis, MO, USA) was used as the curing agent for promoting crosslinking in the epoxy, and was mixed with resin at 1:1 weight ratio. Methylethylketone (MEK, Daejung Chemicals, Siheung-si, Korea) was added to the epoxy resin, as thinner, at 20 mL per 100 g of resin. Ni was selected as the catalyst to promote CNT growth on CF. For Ni electroplating, nickel sulfate hexahydrate (NiSO_4_.6H_2_O) and nickel chloride hexahydrate (NiCl_2_.6H_2_O) were used as the Ni source and were provided by Daejung Chemicals, Siheung-si, Korea. Boric acid (H_3_BO_3_, Daejung Chemicals, Siheung-si, Korea) was used as the pH controller in the plating solution.

### 2.2. Ni Electroplating of CFs

First, fibers were sonicated in acetone (Daejung Chemicals, Siheung-si, Korea) for 12 h to remove the impurities and to desize the thin protective epoxy layer. Second, they were activated in nitric acid for 3 min to increase the oxygen functional groups on the CF surface for better adhesion of the Ni coating layer. Desized and activated CF fabrics were then thoroughly washed with deionized water (DIW) and dried in a vacuum oven at 80 °C for 12 h.

To prepare the plating electrolyte solution, NiSO_4_.6H_2_O and NiCl_2_.6H_2_O were dissolved in DIW at 100 and 40 g/L, respectively. Then, 30 g/L of H_3_BO_3_ was added to the solution to fix the pH at about 4. As-cleaned fibers were then connected to a DC power supply and immersed in the as-prepared solution. The fibers were cathode, and a pure Ni plate (99.99%) was used as the anode. The electrical current at different densities ranging from 15 to 30 A/m^2^, was applied to control the weight of the coated Ni layer. Electroplating was carried out at RT for 1 min. The as-plated fibers were then removed and washed with DIW. Lastly, fibers were heat-treated in a vacuum oven at 200 °C for 30 min to remove the residual salts and moisture.

### 2.3. CVD-Assisted Growth of CNTs

The Ni-coated CFs were put in a zirconia crucible and placed inside a silicon carbide tube. The tube was mounted in an induction heating furnace. The temperature was then increased to 700° at 10 °C/min under 100 cc/min of flowing argon (Ar). When the temperature was stable at 700 °C, Ar flow was stopped, and the carbon source (acetylene gas, C_2_H_2_) was inserted into the chamber at 100 cc/min. The CNT growth process was carried out for 2 min at the target temperature. The chamber was then cooled down to RT at 10 °C/min under 100 cc/min of flowing Ar.

### 2.4. CFRP Fabrication

CFRP samples were prepared by the hand lay-up technique. Specimens were prepared from vacuum-bagged laminates, composed of 3 plies, and cured in a hot-press apparatus at 150 °C under 7.4 MPa for 150 min. The weight fraction of the fillers was about 60 wt.% and 1.8 wt.% of the composite for CFs and MWCNTs, respectively. It should be noted that for the sake of proper comparison, CNT-added CFRP samples were reinforced with Ni-coated CF sheets and MWCNT powder dispersed in the epoxy. As-prepared CFRPs were then cut into desired dimensions (50 × 20 × 1.5 mm^3^) for further analyses using a CNC cutter.

### 2.5. Characterizations

Morphology of the CF samples before and after the CNT modification was investigated via scanning electron microscopy (SEM, JSM-840A, JEOL Ltd., Tokyo, Japan) and transmission electron microscopy (TEM, JEM-2100F, JEOL Ltd., Tokyo, Japan). As-prepared fibers were mixed with epoxy and microtomed prior to the examination by TEM. The crystal structure of the modified CFs with different contents of Ni (10, 15, and 30 wt.%) was analyzed using a wide-angle X-ray diffractometry (XRD, Model D/MAX-III B, Rigaku Corp., Tokyo, Japan) equipped with a rotating anode using CuKα (λ = 0.15418 nm) as the radiation source. The XRD patterns were obtained for the scan range of 10 to 60 degrees at 2°/min. The respective peaks were then indexed and deconvoluted using curve-fitting via the Gaussian function. The work of adhesion (*WA*) between the as-prepared coated fibers and epoxy resin was measured via surface-liquid contact angle measurements. Sessile drop method (Phoenix 300T, Surface Electro Optics SEO Ltd., Suwon-si, Korea) was used to determine the surface energetics of the cured epoxy substrate. The contact angle between a monofilament fiber under tension and two standard liquid probes was measured using a universal force tensiometer (K100SF, Kruss GmbH, Hamburg, Germany). The electrical resistivity of the fabricated CFRPs was measured using a four-probe electrical property analyzer (Loresta-GP MCP-T610, Mitsubishi Chemical Analytech Co., Ltd., Yamato, Japan) according to ASTM-257 and JIS-K6911 standards. As-prepared CFRP samples were also mechanically analyzed using a 3-point bending test (Universal Testing Machine, Instron Model 1125, Instru-Met Corp., Union, NJ, USA) in accordance with ASTM D790 standard. The flexural strength of specimens was analyzed at 0° between CF laminates and the long axis of the sample. The head speed of the load cell was set at 1 mm/min. Ductility index of the composites was also calculated using partial integration of the stress-strain curves after the bending test.

#### 2.5.1. Crystallite Size

The effect of different Ni coatings on the carbon crystallite size (graphitic structure) of CNT-grown samples was studied using the Scherrer equation (Equation (1)) [21]. The full-width at half-maximum (FWHM) and interlayer spacing (d_002_) between the walls of CNTs were calculated via the Origin program (OriginLab Corp., Northampton, MA, USA) curve fitting function.
(1)$$L_{c}=\frac{K\lambda }{\beta \left(2\theta \right)\cdot cos\theta }$$
where *L_c_* is the crystallite size perpendicular to the lattice planes [22]; *K* is the Scherrer constant (*K* = 0.89); *λ* is the X-ray beam wavelength; *β*(2*θ*) is the FWHM.

#### 2.5.2. Surface Energetics

Surface free energy (SFE) of different coated CFs was measured by the sessile drop method using two standard probe liquids with known surface tensions. DIW and diiodomethane (Sigma-Aldrich Chemicals Company, St. Louis, MO, USA) were employed to calculate the polar and dispersive (or London dispersion force) components of SFE. The measurement process was carried out by two main courses of action. First, the surface energetics of epoxy was determined using the following steps:

i.

The contact angles of two probe liquids were individually measured against neat cured-epoxy substrates. To achieve a smooth surface, the substrate was sequentially polished with sandpaper prior to the measurement. Measurement was done several times to ensure the reliability of the analysis.The measured contact angles were then used as inputs for an image analysis program, Surfaceware 9, Surface Electro Optics SEO Ltd., Suwon-si, Korea. Owens-Wendt’s theory [23] was used to determine the polar and dispersive components of SFE of the epoxy samples.

Second, single fibers were drawn out of fabrics using a tweezer under magnifying glass. The following, also mentioned elsewhere [10], describe the steps carried out for SFE measurement of monofilament fibers:

ii.

The weight of the dry fibers was automatically tared by the instrument.Wilhelmy plate method [24] was employed to measure the interfacial tension at the fiber-liquid interface. The filaments were oriented parallel to the gravity axis and partly immersed in the probe liquid. The free end of the fiber in the air was connected to a micro-scale to measure the force required to pull out the fiber from the liquid.Liquid-fiber contact angle and wetted perimeter of fiber surface were measured.Equation (2) (or Equation (3)) was then used to determine the SFE value of monofilaments.

(2)F=W+Pγcosθ−ρgy·A
where *F* is the measured SFE; *P* is the wetted perimeter of the fiber; *γ* is the surface tension of the wetting liquid; *cosθ* is the contact angle at fiber surface; *W* is the weight of the fiber; ρ is the density of the wetting liquid; *g* is the gravitational acceleration; *y* is the immersion depth; *A* is the cross-sectional area of the fiber. The weight of the fiber was tared prior to the measurement, and the buoyancy effect was neglected due to being smaller than the measured force by several orders of magnitude. The simplified form of Equation (2) is shown below:(3)F=Pγcosθ

With the SFE of matrix and filler known, Equation (4) was then used to calculate the work of adhesion between the epoxy and treated fibers. In composites, *WA* is defined as the amount of work required to separate matrix from filler or vice versa. It is represented by energy per unit of surface area or force per unit of length.
(4)$$WA=2{\left(\gamma_{f}^{D}\cdot \gamma_{m}^{D}\right)}^{1/2}+2{\left(\gamma_{f}^{P}\cdot \gamma_{m}^{P}\right)}^{1/2}$$
where *f* and *m* subscripts stand for filler and matrix, respectively. Superscripts *D* and *P* represent dispersive and polar components of SFE, respectively.

## 3. Results and Discussion

### 3.1. Morphology of As-Prepared CFs

Formation of Ni catalyst and CVD-grown CNT bundles was confirmed through SEM images, and can be seen in Figure 1. SEM images of the CNT-grafted CFs with different weight ratios of the Ni-layer are shown in Figure 2. Different magnifications were used to clearly observe the effect of Ni concentration on CNT growth. Comparing different CNT-grafted cases, the development of a multi-phase carbon structure was observed. In Figure 2d, the cylindrical structures were not CNTs but carbonaceous fibers or fibrous carbon clusters. In Figure 2f, it can be clearly seen that these structures were converting to CNTs and were appearing in much smaller diameters. In the case of Figure 2g,h, they were the mixture of CNTs (crystalline phase) and carbon clusters (amorphous phase). The presence of a larger amount of nickel catalysts caused the formation of not only CNTs but also carbonaceous structures, which were hard to define as they were mixed structures. This can be attributed to the non-uniformity of the Ni-plated zones and weak dispersion of the Ni particles on CF. Moreover, higher Ni concentration led to the larger agglomeration of carbon nucleation zones, causing the formation of a hihgly amorphous phase.

The crystal structure of as-coated samples was studied by XRD. The respective diffractograms are shown in Figure 3. 002 and 101 carbon lattices and the representative miller indices of face-centered cube (FCC) Ni lattice can be seen in Figure 3a. It should be noted that 002 and 101 indices represented the lattices oriented perpendicular and parallel to the graphene layers in the graphitic crystal structure, respectively. The XRD pattern of CNT-coated fibers shows the CNT integrated peaks at 43 and 44.5°. It can be seen in Figure 3b that Ni peaks were broadened after the CNT growth, which was attributed to the disruption of crystal structure in the Ni layer, implying the tip-growth formation of CNTs. This confirmed the dominancy of tip-grown CNTs on carbon substrates as explained by Kim et al. [25]. It can be seen that the as-received CF diffractogram is appearing slightly different in Figure 3a,b due to the different normalization of patterns. The shoulder peak near the 002 diffracted peak can be the result of the present turbostratic structure within the CF. Moreover, Figure 3c,d illustrate the deconvoluted peaks for CVD-grown CNTs. As a more amorphous portion results in more broadening of the diffraction peaks, it can be clearly seen that CF/CNT-Ni15 shows the least broadening effect, implying that a higher crystalline portion was obtained in this case.

The values calculated for the carbon crystallite size are shown in Table 1. Ideally, the crystallite size represents the distance between the most inner wall and the most outer wall of the CNT [26]. However, due to the presence of a multi-phase structure, i.e. a mixture of CNTs, fibrous carbon, and CF substrate, in the actual samples, the crystallite size measured here is describing the height of the graphitic lattice present in the cluster. Moreover, according to the findings of [25], higher crystallite size of CNTs can be attributed to stronger wrappings of the graphitic layer on the CF surface around the Ni particles. This can be promoted with better dispersion of catalytic particles on the substrate. According to the results shown in Table 1, it can be deduced that Ni at 15 wt.% resulted in the largest graphitic crystallite size grown via CVD.

### 3.2. Surface Energetics

Table 2 represents the values for each SFE component as well as calculated *WA*. As seen here, CNT-grown CF with 15 wt.% of Ni catalyst shows the highest work of adhesion. Normally, larger crystallite size in the graphitic structure results in better physical properties, such as surface area. According to the adsorption physics, the dispersive component of SFE is highly dependent on the specific surface area. This means that, based on the findings, the larger surface area was obtained for samples containing 15 wt.% Ni. This was also confirmed via the SEM images shown in Figure 2. Considering the aforementioned correlation between the crystallite size and surface energetics, results shown in Table 2 conform well with the findings for carbon crystallite size shown in Table 1. Therefore, 15 wt.% Ni was found to be the optimum weight fraction for CVD-assisted grafting of CNTs on CF. As mentioned before, less amount of Ni can lead to poorly dispersed CNT nucleation zones, where higher Ni concentration can promote CNT agglomeration zones and CF surface degradation due to the high intensity of the electroplating process.

Figure 4 shows the TEM images of CNT-coated CF. This indicates how optimized CVD-grown CNTs can enhance the surface energy of CF, in turn improving the matrix-filler interface by promoting the wettability of filler with resin.

### 3.3. CFRP Electrical Resistivity (Specific Electrical Resistance)

The values for electrical resistivity of CFRPs reinforced with different coating conditions are shown in Figure 5. As seen here, CNT enhanced the electrical conductivity with its large number of free electrons and low band-gap. This accorded well with our previous findings in CNT-reinforced composites [27]. Comparing the samples where CNT was grown and just added to the composite, CNT-grown CFRP shows the lowest electrical resistivity. This was attributed to the better interfacial adhesion between the epoxy matrix and CF coated with CVD-grown CNTs. Results show that grafting CNTs on CF dramatically enhanced the electrical transport in CFRPs compared with that of resin reinforced with virgin CFs.

Herein, a maximum reduction of about 809% in electrical resistivity was found compared with that of virgin CFRP. Table 3 summarizes the recent studies on the improvement of electrical conductivity, or reduction in electrical resistivity, in CNT-reinforced CFRPs. It shows that our optimization in catalyst weight fraction led to one of the lowest electrical resistivities reported for CNT-reinforced epoxy-based CFRPs.

### 3.4. CFRP Mechanical Properties

Figure 6 shows the stress-strain (S-S) graphs obtained for CFRP samples after the 3-point bending test. The highest yield strength was obtained for CFRP reinforced with CNT-grown CFs, followed by CNT-added Ni-CFRP, Ni-CFRP, and as-received CFRP, respectively. The calculated flexural strength of the composites, using Equation (5), is shown in Table 4. These findings, again, confirm a well-adhered composite system using optimized CNT-grafted CFs. The formation of CNT agglomeration zones was likelier in the case of CNT-added Ni-CFRP, which can result in higher crack propagation rates within the composite. Table 5 compares the recent studies on the improvement of flexural strength in CNT-reinforced CFRPs.
(5)$$\sigma =\frac{3FL}{2bt^{2}}$$
where *F* is the exerted force; *L* is the sample length; *b* is the sample width; *t* is the thickness of the sample.

Aside from the flexural strength, samples showed different toughness responses under the same conditions. Obtaining a fine balance between toughness and strength has still been a challenge in modern materials science [40]. To better correlate the strength and toughness of samples, the ductility index (*DI*) was measured and compared (see Table 4). *DI* represents the ratio of plasticity to elasticity in composites. In other words, it shows how well a material can absorb energy before going under plastic deformation. The higher is the *DI*, the less brittle is the material. Figure 7 and Equation (6) describe how to calculate the *DI* from the S-S graph. The values interestingly show that CNT dispersed in epoxy (CNT-added Ni-CFRPs) has led to the least brittle response under the exerted load. This effect can be attributed to the lack of disruption in the Ni coating layer, whereas the formation of CNTs through the tip-growth mechanism broke down the ductile Ni layer, leading to lower ductility of the respective CFRPs. Furthermore, these results indicate that the addition of mechanically strong CNTs can also enhance the ductility of CFRPs.
(6)$$DI=\frac{\int PR}{\int IR}$$
where *PR* and *IR* represent the areas of the S-S graph before and after the ultimate load, respectively. These parameters are shown in Figure 7.

## 4. Conclusions

The effect of different Ni coatings and CNT-grafting on carbon crystallite size and surface energetics of CF substrate was studied in detail. Moreover, fabricated CFRPs reinforced with as-received and as-modified CFs were electrically and mechanically analyzed. The main conclusions are summarized and listed below:In this work, Ni at 15 wt.% showed the highest yield of CNTs. This was explained by the correlation between the carbon crystallite size and surface free energy of CNT-grafted CFs. We showed that better purity of the CNT structure and larger graphitic lattice size can be achieved by simply optimizing the catalyst weight ratio. The presence of a multi-phase (crystalline and amorphous) carbonaceous structure, which is unfavorable for obtaining a high surface area, can be minimized using the proposed method.The electrical resistivity of CNT-grown Ni-CFRP was found to be about 9, 7.5, and 2.3 times lower than those of as-received CFRP, Ni-CFRP, and CNT-added Ni-CFRP, respectively.Results of the 3-point bending test show a 52.9% improvement in the flexural strength of CNT-grown Ni-CFRPs compared with that of virgin CFRP samples.The ductility test confirmed that CNT growth was in a tip-growth manner. The highest ductility index was obtained for CNT-added Ni-CFRP, where Ni-CFRP and CNT-grown Ni-CFRP had approximately the same indices. This shows that even mere addition of CNT powder can highly improve the load-bearing performance of CFRPs. Moreover, it should be noted that the CNT aspect ratio can be highly influential in CNT-added Ni-CFRP properties.

Altogether, our results show that the proper dispersion of catalysts can considerably enhance the graphitic crystal growth via CVD. This indicates that unprecedented improvements of properties can be achieved in composites by individual optimization of the treatment steps. The findings of this study are expected to pave the way for better improvements in composite properties via simple modification of the manufacturing techniques.

## Figures and Tables

**Figure 1 polymers-12-01432-f001:**
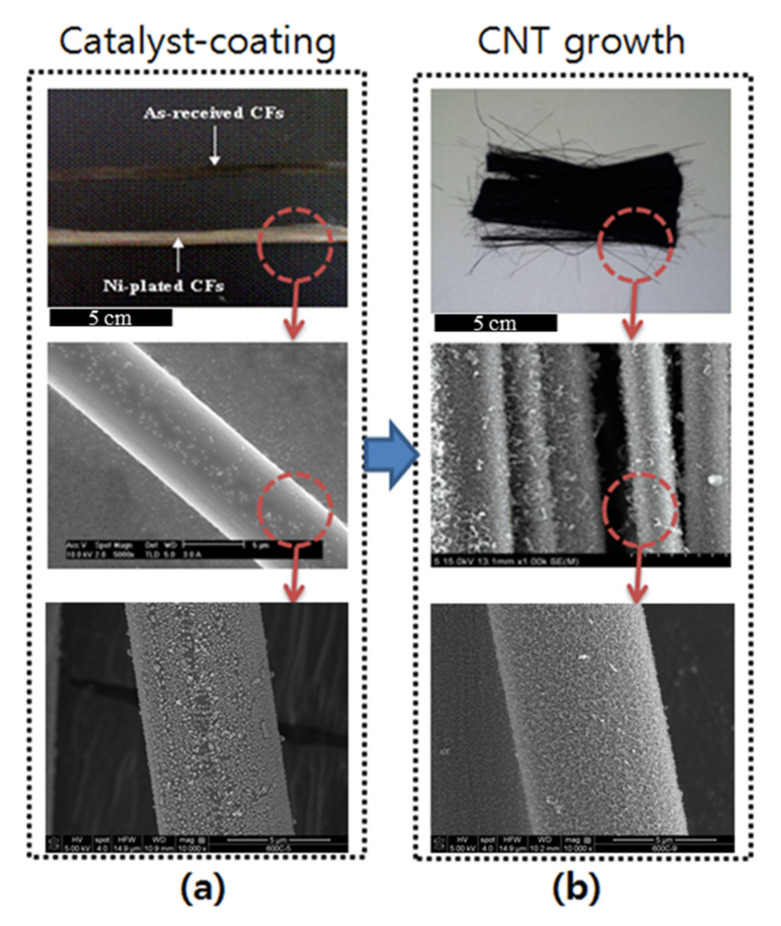
Photographs and SEM images of carbon fibers (CFs); (**a**) after catalyst-coating and (**b**) after CNT growth.

**Figure 2 polymers-12-01432-f002:**
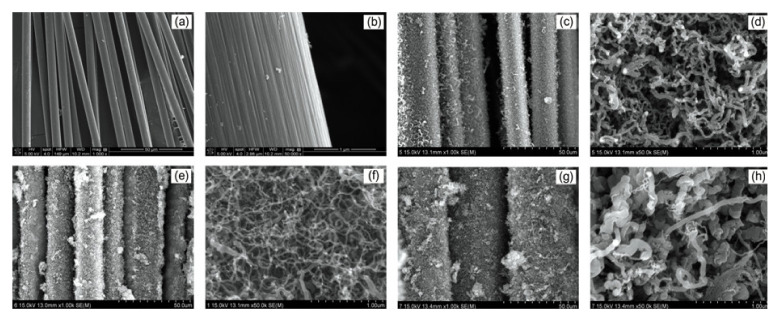
SEM images of CF/CNT hybrids with different catalyst-coating conditions; (**a**) as-received CFs (×1k), (**b**) as-received CFs (×50k), (**c**) CF/CNT-Ni10 (×1k), (**d**) CF/CNT-Ni10 (×50k), (**e**) CF/CNT-Ni15 (×1k), (**f**) CF/CNT-Ni15 (×50k), (**g**) CF/CNT-Ni30 (×1k), (**h**) CF/CNT-Ni30 (×50k).

**Figure 3 polymers-12-01432-f003:**
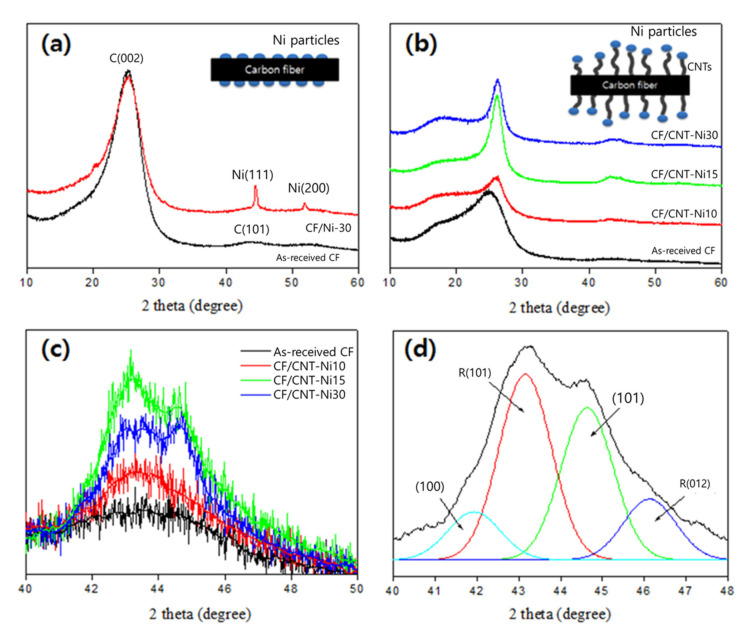
XRD patterns of CF/CNT hybrids as a function of different catalyst-coating conditions; (**a**) comparison between as-received CFs and catalyst-coated CFs (CF/Ni-30), (**b**) patterns with a scan range of 10~60° 2θ, (**c**) patterns with a scan range of 40~50° 2θ, (**d**) sub-peaks of CF/CNT-Ni15 with a scan range of 40~48° 2θ.

**Figure 4 polymers-12-01432-f004:**
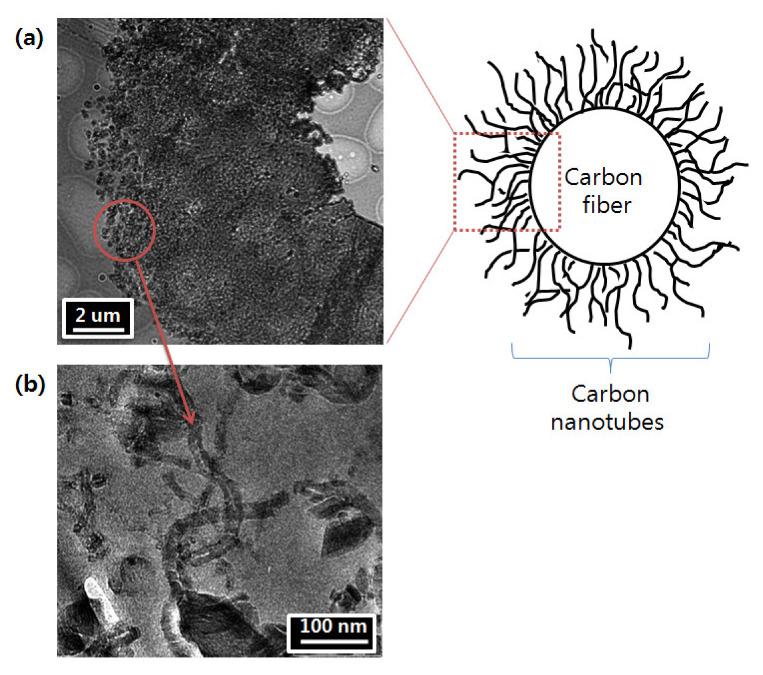
TEM images of CFs after the CNT growth (The CF/CNT hybrid was immersed in epoxy prior to being ultra-microtomed); (**a**) grown CNT cluster; (**b**) magnified image of the CNT structure. The hole observed in (**a**) is the original position of the carbon fiber, which was pulled out during microtoming.

**Figure 5 polymers-12-01432-f005:**
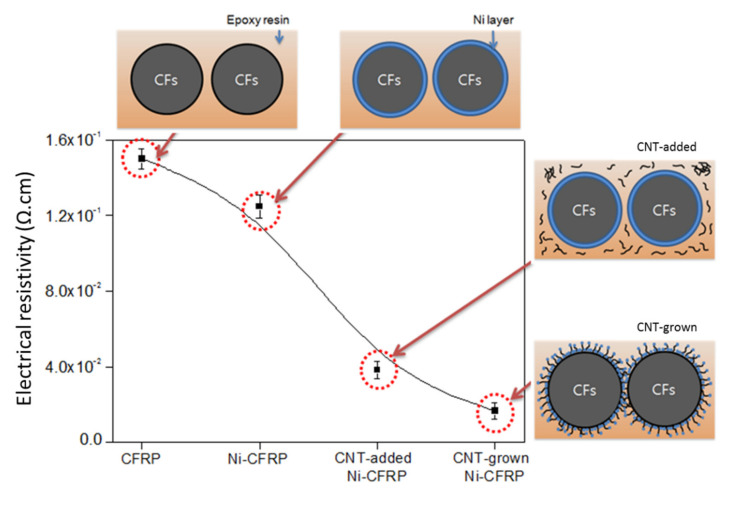
Electrical resistivity of CFRPs with as-received, Ni-plated, CNT-added, and CNT-grown conditions.

**Figure 6 polymers-12-01432-f006:**
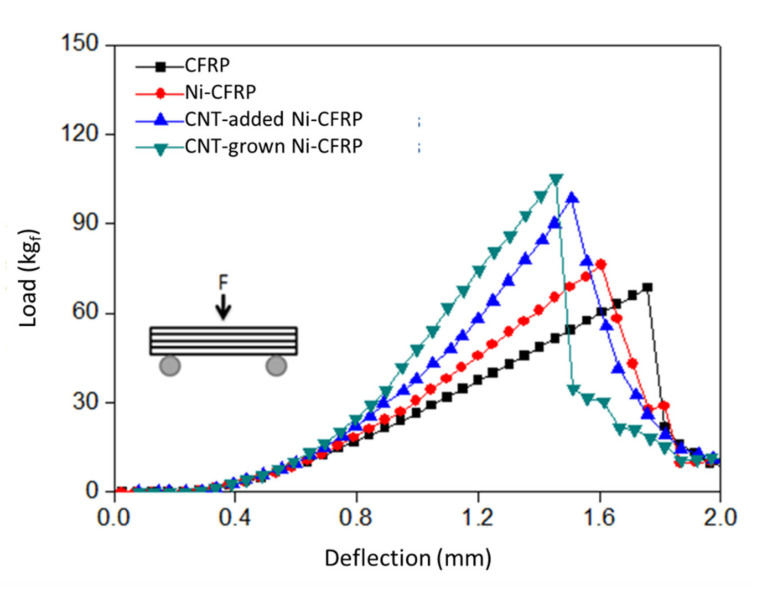
Representative stress-strain curves of CFRPs with as-received, Ni-plated, CNT-added, and CNT-grown conditions.

**Figure 7 polymers-12-01432-f007:**
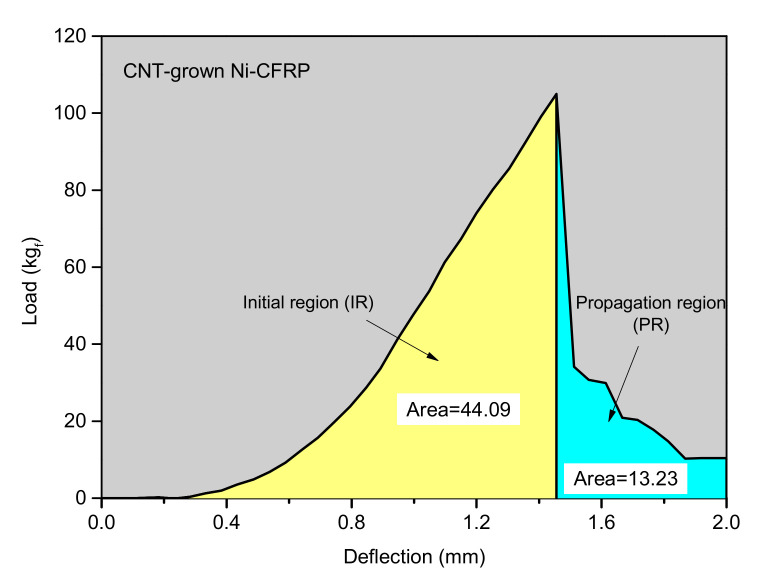
Ductility index (*DI*) measurement technique.

**Table 1 polymers-12-01432-t001:** XRD results of carbon fiber/carbon nanotube (CF/CNT) hybrids as a function of different catalyst-coating conditions.

Sample	2*θ* (°)	d_002_ (nm) ^1^	FWHM (°) ^2^	Crystallite Size (*L_c_*_,_ nm)
As-received CF	24.90	0.357	5.77	1.40
CF/CNT-Ni10	25.95	0.343	3.43	2.35
CF/CNT-Ni15	26.14	0.341	1.77	4.56
CF/CNT-Ni30	26.25	0.339	1.97	4.10

^1^ d_002_ is the interplanar distance between 002 lattice planes. ^2^ FWHM is the full-width at half-maximum intensity (unit: radians).

**Table 2 polymers-12-01432-t002:** Surface energetics of CF/CNT hybrids as a function of different catalyst-coating conditions.

Sample	$\gamma_{f}^{P}\;(\mathrm{mN}/m)$	$\gamma_{f}^{D}\;(\mathrm{mN}/m)$	$\gamma_{f}\;(\mathrm{mN}/m)$	*WA* (mN/m)
As-received CF	3.53	22.58	26.11	76.8
CF/CNT-Ni10	28.52	113.37	141.89	178.9
CF/CNT-Ni15	35.89	140.30	176.19	199.4
CF/CNT-Ni30	13.12	93.62	106.74	155.2

**Table 3 polymers-12-01432-t003:** Literature reports on the electrical enhancement in CNT-reinforced epoxy-based CFRPs compared with virgin CFRP samples.

Study	Reinforcement	Enhancement (%)
Present work	1.8 wt.% MWCNT to prepare 60 wt.% 3-ply CFRPs	809%
[15]	MWCNTs grown on CF fabrics to prepare 2-ply CFRPs	>170
[28]	MWCNTs grown on CF fabrics to prepare x-ply CFRPs	510 in thru-plane direction 330 in in-plane direction
[29]	MWCNTs added at 8 wt.% to prepare 4-ply CFRPs (sanded surface were measured)	789
[30]	MWCNTs added at 20 wt.% to prepare 0.1 wt.% short CFRPs	500
[31]	SWCNTs added at 0.3 wt.% to prepare 12-ply CFRPs	Very high in thru-plane direction (compared with dielectric virgin CFRP) 53.37 in the in-plane direction
[32]	GNP at 0.2 wt.% was added and MWCNTs grown on CFs to prepare 16-ply CFRPs	440
[33]	MWCNTs added at 0.75 wt.% to prepare 60 vol.% CF composites	282
[34]	MWCNTs were grown on CFs to prepare thin-ply CFRPs	654 for epoxy-based CFRPs 2667.6 for polypropylene-based CFRPs
[35]	MWCNTs solution (0.002 g/L) added to prepare 26-ply CFRPs	543
[36]	MWCNT buckypaper sheets added to prepare 8-ply CFRPs	697

**Table 4 polymers-12-01432-t004:** Flexural strength and ductility index (*DI*) of CFRP samples.

Sample	*σ* (MPa)	*DI*
As-received CFRPs	1120.88 ± 3%	0.21 ± 3%
Ni-CFRPs	1246.41 ± 3%	0.29 ± 3%
CNT-added Ni-CFRPs	1612.89 ± 3%	0.42 ± 3%
CNT-grown Ni-CFRPs	1713.89 ± 3%	0.30 ± 3%

**Table 5 polymers-12-01432-t005:** Literature reports on the enhancement of flexural strength in CNT-reinforced CFRPs compared with virgin CFRP samples.

Study	Reinforcement	Enhancement (%)
Present work	1.8 wt.% MWCNT to prepare 60 wt.% 3-ply CFRPs	52.9
[29]	MWCNTs added at 8 wt.% to prepare 4-ply CFRPs	2.7
[31]	SWCNTs added at 0.3 wt.% to prepare 12-ply CFRPs	22.6
[32]	GNP at 0.2 wt.% was added and MWCNTs grown on CFs to prepare 16-ply CFRPs	19
[37]	MWCNTs added at 0.5 wt.% to prepare 8-ply CFRPs	19.7
[38]	MWCNTs grown on CFs to prepare 6-ply CFRPs	5
[39]	MWCNTs grown on CF to prepare CFRP containing 5 wt.% reinforcement	36.7

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
