# Peer review of "Enhanced Surface Energetics of CNT-Grafted Carbon Fibers for Superior Electrical and Mechanical Properties in CFRPs"

_polymers, 2020, doi:10.3390/polym12061432_

Round 1

Reviewer 1 Report

It is requested that the following comments be addressed before the manuscript is considered for publication.

In line 54, the authors cite a disadvantage of chemical treatments of fibers to be the creation of pits, but in line 59, cite their own on work on electrochemical etching as improving composite performance. Does etching not produce pits as well?

In lines 88-90, the authors state measurements of surface energetics to be underreported in the literature. Please explain precisely why that information is required if a given treatment is shown to improve fiber-matrix adhesion?

The authors have chosen to report only surface energies of the fiber and matrix as measures of adhesion. Please clearly describe how these measurements were performed, specifically, how were the fibers prepared?, was the measurement on single fibers or bundles?, how was it performed on the matrix?, etc. Cite the challenges of contact angle measurements on fibers, and describe how they were addressed in your work. Please explain why other measurements such as fiber pull-out tests were not performed.

In lines 242-244, the authors state simply that work of adhesion is directly proportional to the carbon crystallite size, without explaining their reasoning. Please explain.

How do the sizes compare of the CNTs grown on the fibers with those simply added to the mixture? Will that have any influence on composite performance?

In lines 305-309, the lower ductility of the CNT-grown composites is attributed to the breaking up of the nickel layer. How does presence of the nickel layer contribute to ductility?

The meaning of the sentence in lines 321-323 is unclear. What does it mean that optimum weight fractions vary and the results have to be "viewed qualitatively"?

If the CNT-dispersed composites don't show a very different performance in comparison to the CNT-grown composites - what, if any, is the advantage of the CNT graftng step?

In Figure 7, please illustrate how the DI was calculated with an actual load-displacement curve from your measurements, instead of using an idealized drawing.

Please give indications of data variability (standard deviation or 95% confidence interval) in all tabulated results.

Reviewer 2 Report

The manuscript seems to be very interesting and the authors have done a good work to understand the effect of CNT growth on different amount of Ni coating on CF and latter their mechanical properties.

Having said that, need clarity in some of the things mentioned in the manuscript.

From figure 1, it looks like Ni particles are coated around the CF. Figure 1 corresponds to Ni coating of which current density? It looks like uniformly coated around CF. do you have any particle size for Ni? Does the particle size changes with different current densities?

Please put the scale for upper images of figure 1a and 1b.

What is the spaghetti kind of structure in first image of Figure 1b? is it CF or CNT?

Did you observe any kind of alignment in the CNT growth?

What is the yield of CNT in each case?

In connection with the particle size of Ni, the diameter of CNT may also change. Did you observe any trend?

From the SEM image provided in figure 2, it is difficult to make a conclusion that large Ni concentration to more CNT entanglements. Figure 2d and 2f looks not much difference in entanglement, whereas figure 2h looks less entangled in this magnification.

In terms of diameter and size of CNT, figure 2d and 2f seems to be relatively uniform than figure 2h, where different sized (length and diameter) CNT are visible. Any comment on this difference?

Authors have calculated the crystallite size of CNT for different NI concentration catalysts, what actually is crystallite size in CNT?

What about the crystallite size of Ni nanoparticles? Can you calculate it from XRD?

In line 230-233, authors have described the symbols used in eq. 2 and eq. 3, but the symbol used for describing density of the wetting liquid is wrong.

Also, since the symbols used in equation are in italics, use the same font for them while describing too.

The symbols used in Table 2 is a little bit confusing and there are some typographical errors; like WA

What is ‘s’ in this symbol,    ? What is in the table?

Also, if you are writing a letter as subscript, it should look like a subscript. So use the correct tool to write the subscript and superscript.

Figure 4 is not very clear. Authors have mentioned in the caption that ‘The hole observed in (a) is the original position of the carbon fiber which was pulled out during microtoming.’ Is it possible to pull the CF without disturbing the outer CNT layer? The quality of CNT is not as good as, what is shown in figure 2f. Which sample was used for TEM?

Did you do any pre-treatment to CF/CNT in order to mix with uniformly with epoxy?

The figures and results can be explained a little more in details.

Round 2

Reviewer 1 Report

Many thanks for addressing my comments. Please incorporate all your answers in the manuscript text.

Reviewer 2 Report

What is the spaghetti kind of structure in first image of Figure 1b? is it CF or CNT?

The first figure is an optical photograph of the CF bundles. Figure 1a shows the pristine and Ni-coated CF tows. Figure 1b represents the CF bundle after the growth of CNTs.

Is this means that the wire kind of structure projecting from the large bundle is CF coated with Ni and CNT? When you show consecutive images as magnified images, you should be a little more precise on the markers you put on the image to show the magnification.

What is the yield of CNT in each case?

It is arduous to determine the exact amount of only CNT due to the presence of a multiphase carbonaceous structure, i.e. the mixture of crystalline and amorphous carbon. However, based on the SEM analysis of the samples shown in Figure 2, a higher amount of CNT is apparent in the case of samples containing 15 wt.% Ni.

I agree that there are crystalline and amorphous carbonaceous structures present in the as synthesized material. But for the particular synthesis parameters, one can qualitatively calculate the yield of CNT, if you know the weight of catalyst and weight of as synthesized product. Since amorphous carbon is unstable above 350 degree in air, one can find out the amount of crystalline material at 350°. So one can roughly calculate the yield of CNT from all these weight. This is just an information for the authors.

In connection with the particle size of Ni, the diameter of CNT may also change. Did you observe any trend?

It is possible to estimate the number of CNT walls from crystallite parameters considering an ideal nanotube structure. The calculated crystallite size in the manuscript represents the distance between the most inner wall and the most outer wall of multi-walled CNTs [2], which is can act as an approximate indication of CNT radius. However, due to the presence of a multi-phase carbonaceous structure in actual samples, the crystallite size more accurately represents the height of the graphitic lattice present in the mixture, rather than CNT diameter.

What you mean by height of graphitic lattice? There are two XRD patterns shown for as received CF in Figure 3a and Figure 3b. Both looks slightly different from one another, the pattern in Figure 3b has a shoulder peak. Is there any reason? Caption of Figure 3b says a scan range of 10-80° but , actually it is 10-60°. The easy way to calculate the diameter and number of walls of CNT is to take TEM or HRTEM of CNT.

From the SEM image provided in figure 2, it is difficult to make a conclusion that large Ni concentration to more CNT entanglements. Figure 2d and 2f looks not much difference in entanglement, whereas figure 2h looks less entangled in this magnification.

As you can see in Figure 2d and XRD results, they are not CNTs, but carbonaceous fibers or fibrous carbons. It means that they are converting to CNTs, so the mechanical and electrical properties must be very low compared to those of the Figure 2f. In case of Figures 2g and h, they are the mixture of CNTs and some carbon clusters. The presence of large amount of nickel catalysts caused not only CNTs but also carbonaceous structures which are hard to define as they are mixed structures. That’s why the surface free energy of the samples CF/CNT-Ni30 is less than that of CF/CNT-Ni15 sample.

According to the authors, CF/CNT-Ni10 contains carbonaceous fibers or fibrous carbons. What do you mean by carbonaceous fibers or fibrous carbons? Is it amorphous carbon? Because in the XRD of CF/CNT-Ni10, the broad peak around 25° may be due to the amorphous content present in the sample. Whereas the peak at 25° suggest more of crystalline nature for CF/CNT-Ni15 and CF/CNT-Ni30. But, the SEM image displayed in figure 2h suggest that the CNT grown here is not uniform in terms of length and diameter.

Authors have calculated the crystallite size of CNT for different Ni concentration catalysts, what actually is crystallite size in CNT?

There are three parameters normally used for quantifying crystalline structure in CNTs, namely in-plane crystallite size (La), crystallite size perpendicular to planes (Lc), and interplanar spacing (d002). The Figure R1 below, adopted from [2], shows the dimensions defined by these parameters. Scherrer equation can be used for Lc measurement [3]. Please note that the structure in Figure R1 is idealized. The actual samples contain a mixed carbonaceous structure (crystalline and amorphous) and the measured crystallite size represents the height of the graphitic lattice present in the mixture.

Figure 1R. Lattice parameters in an ideal CNT structure (adopted from [3]).

If you really want to calculate the crystallite size, you have to get rid of either amorphous carbon from the material or amorphous broad part from the XRD pattern.
